# Transformers are Sample-Efficient World Models

**Vincent Micheli**[*]
University of Geneva

**Eloi Alonso**[*]
University of Geneva

**François Fleuret**
University of Geneva

## Abstract

Deep reinforcement learning agents are notoriously sample inefficient, which considerably limits their application to real-world problems. Recently, many model-based methods have been designed to address this issue, with learning in the imagination of a world model being one of the most prominent approaches. However, while virtually unlimited interaction with a simulated environment sounds appealing, the world model has to be accurate over extended periods of time. Motivated by the success of Transformers in sequence modeling tasks, we introduce IRIS, a data-efficient agent that learns in a world model composed of a discrete autoencoder and an autoregressive Transformer. With the equivalent of only two hours of gameplay in the Atari 100k benchmark, IRIS achieves a mean human normalized score of 1.046, and outperforms humans on 10 out of 26 games, setting a new state of the art for methods without lookahead search. To foster future research on Transformers and world models for sample-efficient reinforcement learning, we release our code and models at `https://github.com/eloialonso/iris`.

## 1 Introduction

Deep Reinforcement Learning (RL) has become the dominant paradigm for developing competent agents in challenging environments. Most notably, deep RL algorithms have achieved impressive performance in a multitude of arcade (Mnih et al., 2015; Schrittwieser et al., 2020; Hafner et al., 2021), real-time strategy (Vinyals et al., 2019; Berner et al., 2019), board (Silver et al., 2016; 2018; Schrittwieser et al., 2020) and imperfect information (Schmid et al., 2021; Brown et al., 2020a) games. However, a common drawback of these methods is their extremely low sample efficiency. Indeed, experience requirements range from months of gameplay for DreamerV2 (Hafner et al., 2021) in Atari 2600 games (Bellemare et al., 2013b) to thousands of years for OpenAI Five in Dota2 (Berner et al., 2019). While some environments can be sped up for training agents, real-world applications often cannot. Besides, additional cost or safety considerations related to the number of environmental interactions may arise (Yampolskiy, 2018). Hence, sample efficiency is a necessary condition to bridge the gap between research and the deployment of deep RL agents in the wild.

Model-based methods (Sutton & Barto, 2018) constitute a promising direction towards data efficiency. Recently, world models were leveraged in several ways: pure representation learning (Schwarzer et al., 2021), lookahead search (Schrittwieser et al., 2020; Ye et al., 2021), and learning in imagination (Ha & Schmidhuber, 2018; Kaiser et al., 2020; Hafner et al., 2020; 2021). The latter approach is particularly appealing because training an agent inside a world model frees it from sample efficiency constraints. Nevertheless, this framework relies heavily on accurate world models since the policy is purely trained in imagination. In a pioneering work, Ha & Schmidhuber (2018) successfully built imagination-based agents in toy environments. SimPLe recently showed promise in the more challenging Atari 100k benchmark (Kaiser et al., 2020). Currently, the best Atari agent learning in imagination is DreamerV2 (Hafner et al., 2021), although it was developed and evaluated with two hundred million frames available, far from the sample-efficient regime. Therefore, designing new world model architectures, capable of handling visually complex and partially observable environments with few samples, is key to realize their potential as surrogate training grounds.

The Transformer architecture (Vaswani et al., 2017) is now ubiquitous in Natural Language Processing (Devlin et al., 2019; Radford et al., 2019; Brown et al., 2020b; Raffel et al., 2020), and is also gaining traction in Computer Vision (Dosovitskiy et al., 2021; He et al., 2022), as well as in Offline

---

[*]Equal contributions, order determined by a coin flip. Correspondence: `{first.last}@unige.ch`

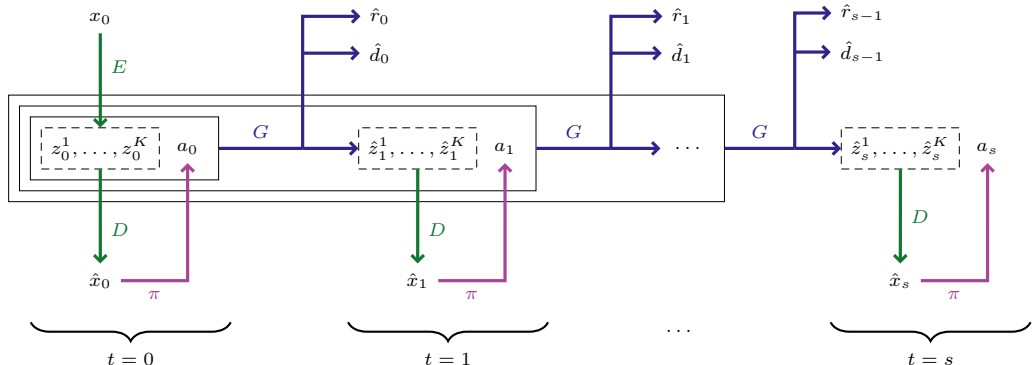

Figure 1: Unrolling imagination over time. This figure shows the policy $\pi$, depicted with purple arrows, taking a sequence of actions in imagination. The green arrows correspond to the encoder $E$ and the decoder $D$ of a discrete autoencoder, whose task is to represent frames in its learnt symbolic language. The backbone $G$ of the world model is a GPT-like Transformer, illustrated with blue arrows. For each action that the policy $\pi$ takes, $G$ simulates the environment dynamics, by autoregressively unfolding new frame tokens that $D$ can decode. $G$ also predicts a reward and a potential episode termination. More specifically, an initial frame $x_0$ is encoded with $E$ into tokens $z_0 = (z_0^1, \ldots, z_0^K) = E(x_0)$. The decoder $D$ reconstructs an image $\hat{x}_0 = D(z_0)$, from which the policy $\pi$ predicts the action $a_0$. From $z_0$ and $a_0$, $G$ predicts the reward $\hat{r}_0$, episode termination $\hat{d}_0 \in \{0, 1\}$, and in an autoregressive manner $\hat{z}_1 = (\hat{z}_1^1, \ldots, \hat{z}_1^K)$, the tokens for the next frame. A dashed box indicates image tokens for a given time step, whereas a solid box represents the input sequence of $G$, *i.e.* $(z_0, a_0)$ at $t = 0$, $(z_0, a_0, \hat{z}_1, a_1)$ at $t = 1$, etc. The policy $\pi$ is purely trained with imagined trajectories, and is only deployed in the real environment to improve the world model $(E, D, G)$.

Reinforcement Learning (Janner et al., 2021; Chen et al., 2021). In particular, the GPT (Radford et al., 2018; 2019; Brown et al., 2020b) family of models delivered impressive results in language understanding tasks. Similarly to world models, these attention-based models are trained with high-dimensional signals and a self-supervised learning objective, thus constituting ideal candidates to simulate an environment.

Transformers particularly shine when they operate over sequences of discrete tokens (Devlin et al., 2019; Brown et al., 2020b). For textual data, there are simple ways (Schuster & Nakajima, 2012; Kudo & Richardson, 2018) to build a vocabulary, but this conversion is not straightforward with images. A naive approach would consist in treating pixels as image tokens, but standard Transformer architectures scale quadratically with sequence length, making this idea computationally intractable. To address this issue, VQGAN (Esser et al., 2021) and DALL-E (Ramesh et al., 2021) employ a discrete autoencoder (Van Den Oord et al., 2017) as a mapping from raw pixels to a much smaller amount of image tokens. Combined with an autoregressive Transformer, these methods demonstrate strong unconditional and conditional image generation capabilities. Such results suggest a new approach to design world models.

In the present work, we introduce IRIS (Imagination with auto-Regression over an Inner Speech), an agent trained in the imagination of a world model composed of a discrete autoencoder and an autoregressive Transformer. IRIS learns behaviors by accurately simulating millions of trajectories. Our approach casts dynamics learning as a sequence modeling problem, where an autoencoder builds a language of image tokens and a Transformer composes that language over time. With minimal tuning, IRIS outperforms a line of recent methods (Kaiser et al., 2020; Hessel et al., 2018; Laskin et al., 2020; Yarats et al., 2021; Schwarzer et al., 2021) for sample-efficient RL in the Atari 100k benchmark (Kaiser et al., 2020). After only two hours of real-time experience, it achieves a mean human normalized score of 1.046, and reaches superhuman performance on 10 out of 26 games. We describe IRIS in Section 2 and present our results in Section 3.

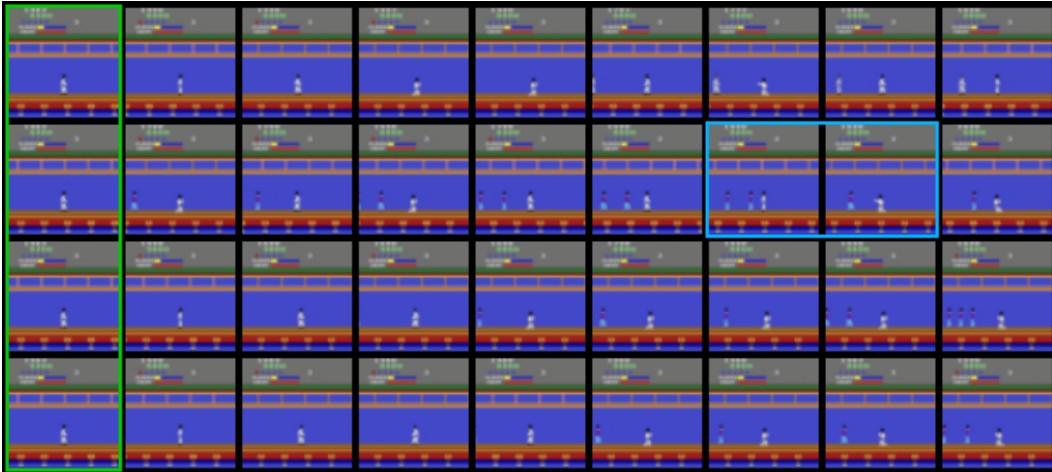

Figure 2: Four imagined trajectories in *KungFuMaster*. We use the same conditioning frame across the four rows, in green, and let the world model imagine the rest. As the initial frame only contains the player, there is no information about the enemies that will come next. Consequently, the world model generates different types and numbers of opponents in each simulation. It is also able to reflect an essential game mechanic, highlighted in the blue box, where the first enemy disappears after getting hit by the player.

## 2 METHOD

We formulate the problem as a Partially Observable Markov Decision Process (POMDP) with image observations $x_t \in \mathbb{R}^{h \times w \times 3}$, discrete actions $a_t \in \{1, \ldots, A\}$, scalar rewards $r_t \in \mathbb{R}$, episode termination $d_t \in \{0, 1\}$, discount factor $\gamma \in (0, 1)$, initial observation distribution $\rho_0$, and environment dynamics $x_{t+1}, r_t, d_t \sim p(x_{t+1}, r_t, d_t \mid x_{\leq t}, a_{\leq t})$. The reinforcement learning objective is to train a policy $\pi$ that yields actions maximizing the expected sum of rewards $\mathbb{E}_\pi[\sum_{t \geq 0} \gamma^t r_t]$.

Our method relies on the three standard components to learn in imagination (Sutton & Barto, 2018): experience collection, world model learning, and behavior learning. In the vein of Ha & Schmidhuber (2018); Kaiser et al. (2020); Hafner et al. (2020; 2021), our agent learns to act exclusively within its world model, and we only make use of real experience to learn the environment dynamics.

We repeatedly perform the three following steps:

- `collect_experience`: gather experience in the real environment with the current policy.

- `update_world_model`: improve rewards, episode ends and next observations predictions.

- `update_behavior`: in imagination, improve the policy and value functions.

The world model is composed of a discrete autoencoder (Van Den Oord et al., 2017), to convert an image to tokens and back, and a GPT-like autoregressive Transformer (Vaswani et al., 2017; Radford et al., 2019; Brown et al., 2020b), whose task is to capture environment dynamics. Figure 1 illustrates the interplay between the policy and these two components during imagination. We first describe the autoencoder and the Transformer in Sections 2.1 and 2.2, respectively. Section 2.3 then details the procedure to learn the policy and value functions in imagination. Appendix A provides a comprehensive description of model architectures and hyperparameters. Algorithm 1 summarizes the training protocol.

### 2.1 FROM IMAGE OBSERVATIONS TO TOKENS

The discrete autoencoder $(E, D)$ learns a symbolic language of its own to represent high-dimensional images as a small number of tokens. The back and forth between frames and tokens is illustrated with green arrows in Figure 1.

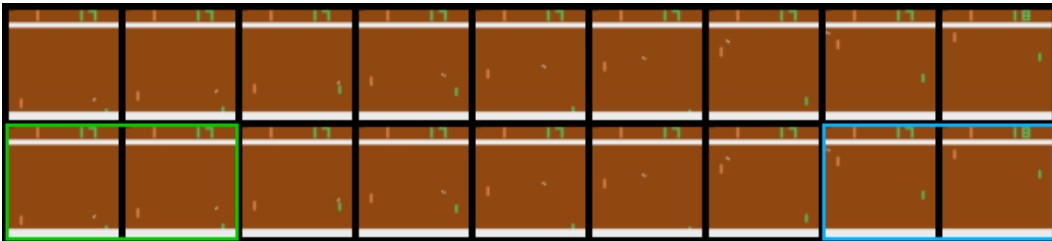

Figure 3: Pixel perfect predictions in *Pong*. The top row displays a test trajectory collected in the real environment. The bottom row depicts the reenactment of that trajectory inside the world model. More precisely, we condition the world model with the first two frames of the true sequence, in green. We then sequentially feed it the true actions and let it imagine the subsequent frames. After only 120 games of training, the world model perfectly simulates the ball's trajectory and players' movements. Notably, it also captures the game mechanic of updating the scoreboard after winning an exchange, as shown in the blue box.

More precisely, the encoder $E : \mathbb{R}^{h \times w \times 3} \to \{1, \dots, N\}^K$ converts an input image $x_t$ into $K$ tokens from a vocabulary of size $N$. Let $\mathcal{E} = \{e_i\}_{i=1}^N \in \mathbb{R}^{N \times d}$ be the corresponding embedding table of $d$-dimensional vectors. The input image $x_t$ is first passed through a Convolutional Neural Network (CNN) (LeCun et al., 1989) producing output $y_t \in \mathbb{R}^{K \times d}$. We then obtain the output tokens $z_t = (z_t^1, \dots, z_t^K) \in \{1, \dots, N\}^K$ as $z_t^k = \operatorname{argmin}_i \left\| y_t^k - e_i \right\|_2$, the index of the closest embedding vector in $\mathcal{E}$ (Van Den Oord et al., 2017; Esser et al., 2021). Conversely, the CNN decoder $D : \{1, \dots, N\}^K \to \mathbb{R}^{h \times w \times 3}$ turns $K$ tokens back into an image.

This discrete autoencoder is trained on previously collected frames, with an equally weighted combination of a $L_1$ reconstruction loss, a commitment loss (Van Den Oord et al., 2017; Esser et al., 2021), and a perceptual loss (Esser et al., 2021; Johnson et al., 2016; Larsen et al., 2016). We use a straight-through estimator (Bengio et al., 2013) to enable backpropagation training.

## 2.2 MODELING DYNAMICS

At a high level, the Transformer $G$ captures the environment dynamics by modeling the language of the discrete autoencoder over time. Its central role of unfolding imagination is highlighted with the blue arrows in Figure 1.

Specifically, $G$ operates over sequences of interleaved frame and action tokens. An input sequence $(z_0^1, \dots, z_0^K, a_0, z_1^1, \dots, z_1^K, a_1, \dots, z_t^1, \dots, z_t^K, a_t)$ is obtained from the raw sequence $(x_0, a_0, x_1, a_1, \dots, x_t, a_t)$ by encoding the frames with $E$, as described in Section 2.1.

At each time step $t$, the Transformer models the three following distributions:

$$\text{Transition:} \quad \hat{z}_{t+1} \sim p_G\big(\hat{z}_{t+1} \mid z_{\leq t}, a_{\leq t}\big) \text{ with } \hat{z}_{t+1}^k \sim p_G\big(\hat{z}_{t+1}^k \mid z_{\leq t}, a_{\leq t}, z_{t+1}^{<k}\big) \qquad (1)$$

$$\text{Reward:} \quad \hat{r}_t \sim p_G\big(\ \hat{r}_t \ \mid z_{\leq t}, a_{\leq t}\big) \qquad (2)$$

$$\text{Termination:} \quad \hat{d}_t \sim p_G\big(\ \hat{d}_t \ \mid z_{\leq t}, a_{\leq t}\big) \qquad (3)$$

Note that the conditioning for the $k$-th token also includes $z_{t+1}^{<k} := (z_{t+1}^1, \dots, z_{t+1}^{k-1})$, the tokens that were already predicted, i.e. the autoregressive process happens at the token level.

We train $G$ in a self-supervised manner on segments of $L$ time steps, sampled from past experience. We use a cross-entropy loss for the transition and termination predictors, and a mean-squared error loss or a cross-entropy loss for the reward predictor, depending on the reward function.

## 2.3 LEARNING IN IMAGINATION

Together, the discrete autoencoder $(E, D)$ and the Transformer $G$ form a world model, capable of imagination. The policy $\pi$, depicted with purple arrows in Figure 1, exclusively learns in this imagination MDP.

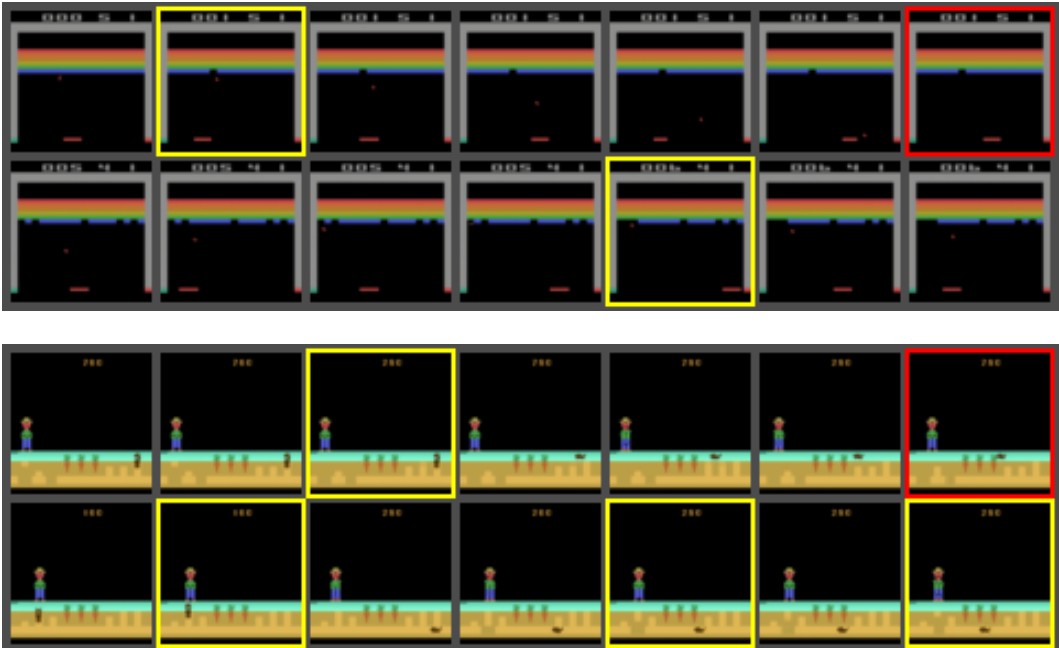

Figure 4: Imagining rewards and episode ends in *Breakout* (top) and *Gopher* (bottom). Each row depicts an imagined trajectory initialized with a single frame from the real environment. Yellow boxes indicate frames where the world model predicts a positive reward. In *Breakout*, it captures that breaking a brick yields rewards, and the brick is correctly removed from the following frames. In *Gopher*, the player has to protect the carrots from rodents. The world model successfully internalizes that plugging a hole or killing an enemy leads to rewards. Predicted episode terminations are highlighted with red boxes. The world model accurately reflects that missing the ball in *Breakout*, or letting an enemy reach the carrots in *Gopher*, will result in the end of an episode.

At time step $t$, the policy observes a reconstructed image observation $\hat{x}_t$ and samples action $a_t \sim \pi(a_t|\hat{x}_{\leq t})$. The world model then predicts the reward $\hat{r}_t$, the episode end $\hat{d}_t$, and the next observation $\hat{x}_{t+1} = D(\hat{z}_{t+1})$, with $\hat{z}_{t+1} \sim p_G(\hat{z}_{t+1} \mid z_0, a_0, \hat{z}_1, a_1, \ldots, \hat{z}_t, a_t)$. This imagination procedure is initialized with a real observation $x_0$ sampled from past experience, and is rolled out for $H$ steps, the imagination horizon hyperparameter. We stop if an episode end is predicted before reaching the horizon. Figure 1 illustrates the imagination procedure.

As we roll out imagination for a fixed number of steps, we cannot simply use a Monte Carlo estimate for the expected return. Hence, to bootstrap the rewards that the agent would get beyond a given time step, we have a value network $V$ that estimates $V(\hat{x}_t) \simeq \mathbb{E}_\pi \left[ \sum_{\tau \geq t} \gamma^{\tau-t} \hat{r}_\tau \right]$.

Many actor-critic methods could be employed to train $\pi$ and $V$ in imagination (Sutton & Barto, 2018; Kaiser et al., 2020; Hafner et al., 2020). For the sake of simplicity, we opt for the learning objectives and hyperparameters of DreamerV2 (Hafner et al., 2021), that delivered strong performance in Atari games. Appendix B gives a detailed breakdown of the reinforcement learning objectives.

## 3 EXPERIMENTS

Sample-efficient reinforcement learning is a growing field with multiple benchmarks in complex visual environments (Hafner, 2022; Kanervisto et al., 2022). In this work, we focus on the well established Atari 100k benchmark (Kaiser et al., 2020). We present the benchmark and its baselines in Section 3.1. We describe the evaluation protocol and discuss the results in Section 3.2. Qualitative examples of the world model's capabilities are given in Section 3.3.

Table 1: Returns on the 26 games of Atari 100k after 2 hours of real-time experience, and human-normalized aggregate metrics. Bold numbers indicate the top methods without lookahead search while underlined numbers specify the overall best methods. IRIS outperforms learning-only methods in terms of number of superhuman games, mean, interquartile mean (IQM), and optimality gap.

| Game | Random | Human | Lookahead search | | No lookahead search | | | | |
| | | | MuZero | EfficientZero | SimPLe | CURL | DrQ | SPR | IRIS (ours) |
|---|---|---|---|---|---|---|---|---|---|
| Alien | 227.8 | 7127.7 | 530.0 | 808.5 | 616.9 | 711.0 | **865.2** | 841.9 | 420.0 |
| Amidar | 5.8 | 1719.5 | 38.8 | 148.6 | 74.3 | 113.7 | 137.8 | **179.7** | 143.0 |
| Assault | 222.4 | 742.0 | 500.1 | 1263.1 | 527.2 | 500.9 | 579.6 | 565.6 | **1524.4** |
| Asterix | 210.0 | 8503.3 | 1734.0 | 25557.8 | **1128.3** | 567.2 | 763.6 | 962.5 | 853.6 |
| BankHeist | 14.2 | 753.1 | 192.5 | 351.0 | 34.2 | 65.3 | 232.9 | **345.4** | 53.1 |
| BattleZone | 2360.0 | 37187.5 | 7687.5 | 13871.2 | 4031.2 | 8997.8 | 10165.3 | **14834.1** | 13074.0 |
| Boxing | 0.1 | 12.1 | 15.1 | 52.7 | 7.8 | 0.9 | 9.0 | 35.7 | **70.1** |
| Breakout | 1.7 | 30.5 | 48.0 | 414.1 | 16.4 | 2.6 | 19.8 | 19.6 | **83.7** |
| ChopperCommand | 811.0 | 7387.8 | 1350.0 | 1117.3 | 979.4 | 783.5 | 844.6 | 946.3 | **1565.0** |
| CrazyClimber | 10780.5 | 35829.4 | 56937.0 | 83940.2 | **62583.6** | 9154.4 | 21539.0 | 36700.5 | 59324.2 |
| DemonAttack | 152.1 | 1971.0 | 3527.0 | 13003.9 | 208.1 | 646.5 | 1321.5 | 517.6 | **2034.4** |
| Freeway | 0.0 | 29.6 | 21.8 | 21.8 | 16.7 | 28.3 | 20.3 | 19.3 | **31.1** |
| Frostbite | 65.2 | 4334.7 | 255.0 | 296.3 | 236.9 | **1226.5** | 1014.2 | 1170.7 | 259.1 |
| Gopher | 257.6 | 2412.5 | 1256.0 | 3260.3 | 596.8 | 400.9 | 621.6 | 660.6 | **2236.1** |
| Hero | 1027.0 | 30826.4 | 3095.0 | 9315.9 | 2656.6 | 4987.7 | 4167.9 | 5858.6 | **7037.4** |
| Jamesbond | 29.0 | 302.8 | 87.5 | 517.0 | 100.5 | 331.0 | 349.1 | 366.5 | **462.7** |
| Kangaroo | 52.0 | 3035.0 | 62.5 | 724.1 | 51.2 | 740.2 | 1088.4 | **3617.4** | 838.2 |
| Krull | 1598.0 | 2665.5 | 4890.8 | 5663.3 | 2204.8 | 3049.2 | 4402.1 | 3681.6 | **6616.4** |
| KungFuMaster | 258.5 | 22736.3 | 18813.0 | 30944.8 | 14862.5 | 8155.6 | 11467.4 | 14783.2 | **21759.8** |
| MsPacman | 307.3 | 6951.6 | 1265.6 | 1281.2 | **1480.0** | 1064.0 | 1218.1 | 1318.4 | 999.1 |
| Pong | -20.7 | 14.6 | -6.7 | 20.1 | 12.8 | -18.5 | -9.1 | -5.4 | **14.6** |
| PrivateEye | 24.9 | 69571.3 | 56.3 | 96.7 | 35.0 | 81.9 | 3.5 | 86.0 | **100.0** |
| Qbert | 163.9 | 13455.0 | 3952.0 | 13781.9 | 1288.8 | 727.0 | **1810.7** | 866.3 | 745.7 |
| RoadRunner | 11.5 | 7845.0 | 2500.0 | 17751.3 | 5640.6 | 5006.1 | 11211.4 | **12213.1** | 9614.6 |
| Seaquest | 68.4 | 42054.7 | 208.0 | 1100.2 | **683.3** | 315.2 | 352.3 | 558.1 | 661.3 |
| UpNDown | 533.4 | 11693.2 | 2896.9 | 17264.2 | 3350.3 | 2646.4 | 4324.5 | **10859.2** | 3546.2 |
| #Superhuman (↑) | 0 | N/A | 5 | 14 | 1 | 2 | 3 | 6 | **10** |
| Mean (↑) | 0.000 | 1.000 | 0.562 | 1.943 | 0.332 | 0.261 | 0.465 | 0.616 | **1.046** |
| Median (↑) | 0.000 | 1.000 | 0.227 | 1.090 | 0.134 | 0.092 | 0.313 | **0.396** | 0.289 |
| IQM (↑) | 0.000 | 1.000 | N/A | N/A | 0.130 | 0.113 | 0.280 | 0.337 | **0.501** |
| Optimality Gap (↓) | 1.000 | 0.000 | N/A | N/A | 0.729 | 0.768 | 0.631 | 0.577 | **0.512** |

## 3.1 BENCHMARK AND BASELINES

Atari 100k consists of 26 Atari games (Bellemare et al., 2013a) with various mechanics, evaluating a wide range of agent capabilities. In this benchmark, an agent is only allowed 100k actions in each environment. This constraint is roughly equivalent to 2 hours of human gameplay. By way of comparison, unconstrained Atari agents are usually trained for 50 million steps, a 500 fold increase in experience.

Multiple baselines were compared on the Atari 100k benchmark. SimPLe (Kaiser et al., 2020) trains a policy with PPO (Schulman et al., 2017) in a video generation model. CURL (Laskin et al., 2020) develops off-policy agents from high-level image features obtained with contrastive learning. DrQ (Yarats et al., 2021) augments input images and averages Q-value estimates over several transformations. SPR (Schwarzer et al., 2021) enforces consistent representations of input images across augmented views and neighbouring time steps. The aforementioned baselines carry additional techniques to improve performance, such as prioritized experience replay (Schaul et al., 2016), epsilon-greedy scheduling, or data augmentation.

We make a distinction between methods with and without lookahead search. Indeed, algorithms relying on search at decision time (Silver et al., 2016; 2018; Schrittwieser et al., 2020) can vastly improve agent performance, but they come at a premium in computational resources and code complexity. MuZero (Schrittwieser et al., 2020) and EfficientZero (Ye et al., 2021) are the current standard for search-based methods in Atari 100k. MuZero leverages Monte Carlo Tree Search (MCTS) (Kocsis & Szepesvári, 2006; Coulom, 2007) as a policy improvement operator, by unrolling multiple hypothetical trajectories in the latent space of a world model. EfficientZero improves upon MuZero by introducing a self-supervised consistency loss, predicting returns over short horizons in one shot, and correcting off-policy trajectories with its world model.

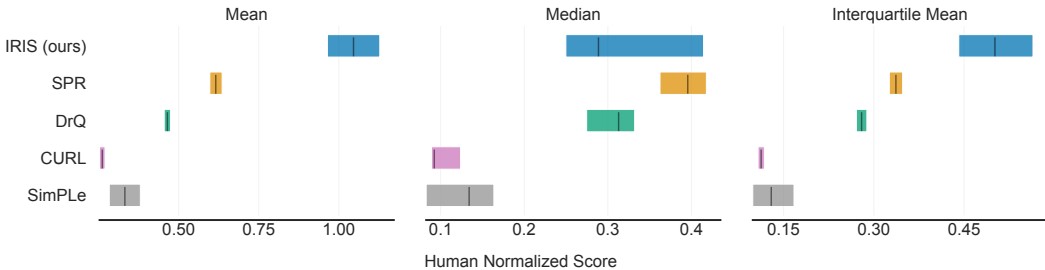

Figure 5: Mean, median, and interquartile mean human normalized scores, computed with stratified bootstrap confidence intervals. 5 runs for IRIS and SimPLe, 100 runs for SPR, CURL, and DrQ (Agarwal et al., 2021).

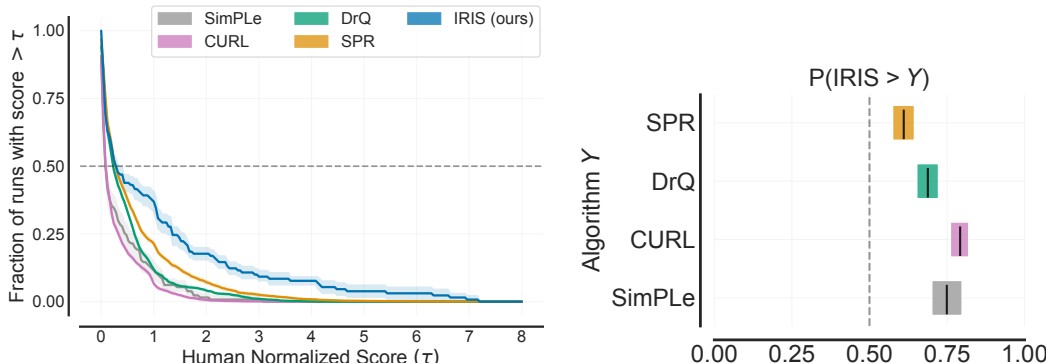

(a) Performance profiles, i.e. fraction of runs above a given human normalized score.

(b) Probabilities of improvement, i.e. how likely it is for IRIS to outperform baselines on any game.

Figure 6: Performance profiles (left) and probabilities of improvement (right) (Agarwal et al., 2021).

## 3.2 RESULTS

The human normalized score is the established measure of performance in Atari 100k. It is defined as $\frac{score\_agent-score\_random}{score\_human-score\_random}$, where *score_random* comes from a random policy, and *score_human* is obtained from human players (Wang et al., 2016).

Table 1 displays returns across games and human-normalized aggregate metrics. For MuZero and EfficientZero, we report the averaged results published by Ye et al. (2021) (3 runs). We use results from the Atari 100k case study conducted by Agarwal et al. (2021) for the other baselines (100 new runs for CURL, DrQ, SPR, and 5 existing runs for SimPLe). Finally, we evaluate IRIS by computing an average over 100 episodes collected at the end of training for each game (5 runs).

Agarwal et al. (2021) discuss the limitations of mean and median scores, and show that substantial discrepancies arise between standard point estimates and interval estimates in RL benchmarks. Following their recommendations, we summarize in Figure 5 the human normalized scores with stratified bootstrap confidence intervals for mean, median, and interquartile mean (IQM). For finer comparisons, we also provide performance profiles and probabilities of improvement in Figure 6.

With the equivalent of only two hours of gameplay, IRIS achieves a superhuman mean score of 1.046 (+70%), an IQM of 0.501 (+49%), an optimality gap of 0.512 (+11%), and outperforms human players on 10 out of 26 games (+67%), where the relative improvements are computed with respect to SPR (Schwarzer et al., 2021). These results constitute a new state of the art for methods without lookahead search in the Atari 100k benchmark. We also note that IRIS outperforms MuZero, although the latter was not designed for the sample-efficient regime.

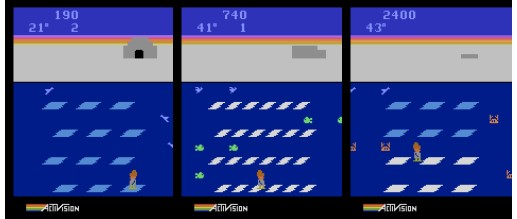 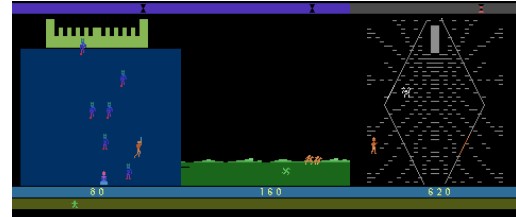

Figure 7: Three consecutive levels in the games *Frostbite* (left) and *Krull* (right). In our experiments, the world model struggles to simulate subsequent levels in *Frostbite*, but not in *Krull*. Indeed, exiting the first level in *Frostbite* requires a long and unlikely sequence of actions to first build the igloo, and then go back to it from the bottom of the screen. Such rare events prevent the world model from internalizing new aspects of the game, which will therefore not be experienced by the policy in imagination. While *Krull* features more diverse levels, the world model successfully reflects this variety, and IRIS even sets a new state of the art in this environment. This is likely due to more frequent transitions from one stage to the next in *Krull*, resulting in a sufficient coverage of each level.

In addition, performance profiles (Figure 6a) reveal that IRIS is on par with the strongest baselines for its bottom 50% of games, at which point it stochastically dominates (Agarwal et al., 2021; Dror et al., 2019) the other methods. Similarly, the probability of improvement is greater than 0.5 for all baselines (Figure 6b).

In terms of median score, IRIS overlaps with other methods (Figure 5). Interestingly, Schwarzer et al. (2021) note that the median is only influenced by a few decisive games, as evidenced by the width of the confidence intervals for median scores, even with 100 runs for DrQ, CURL and SPR.

We observe that IRIS is particularly strong in games that do not suffer from distributional shifts as the training progresses. Examples of such games include *Pong*, *Breakout*, and *Boxing*. On the contrary, the agent struggles when a new level or game mechanic is unlocked through an unlikely event. This sheds light on a double exploration problem. IRIS has to first discover a new aspect of the game for its world model to internalize it. Only then may the policy rediscover and exploit it. Figure 7 details this phenomenon in *Frostbite* and *Krull*, two games with multiple levels. In summary, as long as transitions between levels do not depend on low-probability events, the double exploration problem does not hinder performance.

Another kind of games difficult to simulate are visually challenging environments where capturing small details is important. As discussed in Appendix E, increasing the number of tokens to encode frames improves performance, albeit at the cost of increased computation.

## 3.3 WORLD MODEL ANALYSIS

As IRIS learns behaviors entirely in its imagination, the quality of the world model is the cornerstone of our approach. For instance, it is key that the discrete autoencoder correctly reconstructs elements like a ball, a player, or an enemy. Similarly, the potential inability of the Transformer to capture important game mechanics, like reward attribution or episode termination, can severely hamper the agent's performance. Hence, no matter the amount of imagined trajectories, the agent will learn suboptimal policies if the world model is flawed.

While Section 3.2 provides a quantitative evaluation, we aim to complement the analysis with qualitative examples of the abilities of the world model. Figure 2 shows the generation of many plausible futures in the face of uncertainty. Figure 3 depicts pixel-perfect predictions in *Pong*. Finally, we illustrate in Figure 4 predictions for rewards and episode terminations, which are crucial to the reinforcement learning objective.

## 4 RELATED WORK

### LEARNING IN THE IMAGINATION OF WORLD MODELS

The idea of training policies in a learnt model of the world was first investigated in tabular environments (Sutton & Barto, 2018). Ha & Schmidhuber (2018) showed that simple visual environments could be simulated with autoencoders and recurrent networks. SimPLe (Kaiser et al., 2020) demonstrated that a PPO policy (Schulman et al., 2017) trained in a video prediction model outperformed humans in some Atari games. Improving upon Dreamer (Hafner et al., 2020), DreamerV2 (Hafner et al., 2021) was the first agent learning in imagination to achieve human-level performance in the Atari 50M benchmark. Its world model combines a convolutional autoencoder with a recurrent state-space model (RSSM) (Hafner et al., 2019) for latent dynamics learning. More recently, Chen et al. (2022) explored a variant of DreamerV2 where a Transformer replaces the recurrent network in the RSSM and Seo et al. (2022) enhance DreamerV2 in the setting where an offline dataset of videos is available for pretraining.

### REINFORCEMENT LEARNING WITH TRANSFORMERS

Following spectacular advances in natural language processing (Manning & Goldie, 2022), the reinforcement learning community has recently stepped into the realm of Transformers. Parisotto et al. (2020) make the observation that the standard Transformer architecture is difficult to optimize with RL objectives. The authors propose to replace residual connections by gating layers to stabilize the learning procedure. Our world model does not require such modifications, which is most likely due to its self-supervised learning objective. The Trajectory Transformer (Janner et al., 2021) and the Decision Transformer (Chen et al., 2021) represent offline trajectories as a static dataset of sequences, and the Online Decision Transformer (Zheng et al., 2022) extends the latter to the online setting. The Trajectory Transformer is trained to predict future returns, states and actions. At inference time, it can thus plan for the optimal action with a reward-driven beam search, yet the approach is limited to low-dimensional states. On the contrary, Decision Transformers can handle image inputs but cannot be easily extended as world models. Ozair et al. (2021) introduce an offline variant of MuZero (Schrittwieser et al., 2020) capable of handling stochastic environments by performing an hybrid search with a Transformer over both actions and trajectory-level discrete latent variables.

### VIDEO GENERATION WITH DISCRETE AUTOENCODERS AND TRANSFORMERS

VQGAN (Esser et al., 2021) and DALL-E (Ramesh et al., 2021) use discrete autoencoders to compress a frame into a small sequence of tokens, that a transformer can then model autoregressively. Other works extend the approach to video generation. GODIVA (Wu et al., 2021) models sequences of frames instead of a single frame for text conditional video generation. VideoGPT (Yan et al., 2021) introduces video-level discrete autoencoders, and Transformers with spatial and temporal attention patterns, for unconditional and action conditional video generation.

## 5 CONCLUSION

We introduced IRIS, an agent that learns purely in the imagination of a world model composed of a discrete autoencoder and an autoregressive Transformer. IRIS sets a new state of the art in the Atari 100k benchmark for methods without lookahead search. We showed that its world model acquires a deep understanding of game mechanics, resulting in pixel perfect predictions in some games. We also illustrated the generative capabilities of the world model, providing a rich gameplay experience when training in imagination. Ultimately, with minimal tuning compared to existing battle-hardened agents, IRIS opens a new path towards efficiently solving complex environments.

In the future, IRIS could be scaled up to computationally demanding and challenging tasks that would benefit from the speed of its world model. Besides, its policy currently learns from reconstructed frames, but it could probably leverage the internal representations of the world model. Another exciting avenue of research would be to combine learning in imagination with MCTS. Indeed, both approaches deliver impressive results, and their contributions to agent performance might be complementary.

REPRODUCIBILITY STATEMENT

The different components and their training objectives are introduced in Section 2 and Appendix B. We describe model architectures and list hyperparameters in Appendix A. We specify the resources used to produce our results in Appendix G. Algorithm 1 makes explicit the interplay between components in the training loop. In Section 3.2, we provide the source of the reported results for the baselines, as well as the evaluation protocol.

The code is part of the supplementary materials, and will be open-sourced to ensure reproducible results and foster future research. Minimal dependencies are required to run the codebase and we provide a thorough user guide to get started. Training and evaluation can be launched with simple commands, customization is possible with configuration files, and we include scripts to visualize agents playing and let users interact with the world model.

ETHICS STATEMENT

The development of autonomous agents for real-world environments raises many safety and environmental concerns. During its training period, an agent may cause serious harm to individuals and damage its surroundings. It is our belief that learning in the imagination of world models greatly reduces the risks associated with training new autonomous agents. Indeed, in this work, we propose a world model architecture capable of accurately modeling environments with very few samples. However, in a future line of research, one could go one step further and leverage existing data to eliminate the necessity of interacting with the real world.

ACKNOWLEDGMENTS

We would like to thank Maxim Peter, Bálint Máté, Daniele Paliotta, Atul Sinha, and Alexandre Dupuis for insightful discussions and comments. Vincent Micheli was supported by the Swiss National Science Foundation under grant number FNS-187494.

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

# A  MODELS AND HYPERPARAMETERS

## A.1  DISCRETE AUTOENCODER

Our discrete autoencoder is based on the implementation of VQGAN (Esser et al., 2021). We removed the discriminator, essentially turning the VQGAN into a vanilla VQVAE (Van Den Oord et al., 2017) with an additional perceptual loss (Johnson et al., 2016; Larsen et al., 2016).

The training objective is the following:

$$\mathcal{L}(E, D, \mathcal{E}) = \left\| x - D(z) \right\|_1 + \left\| \text{sg}(E(x)) - \mathcal{E}(z) \right\|_2^2 + \left\| \text{sg}(\mathcal{E}(z)) - E(x) \right\|_2^2 + \mathcal{L}_{perceptual}(x, D(z))$$

Here, the first term is the reconstruction loss, the next two terms constitute the commitment loss (where $\text{sg}(\cdot)$ is the stop-gradient operator), and the last term is the perceptual loss.

Table 2: Encoder / Decoder hyperparameters. We list the hyperparameters for the encoder, the same ones apply for the decoder.

| Hyperparameter | Value |
| --- | --- |
| Frame dimensions (h, w) | $64 \times 64$ |
| Layers | 4 |
| Residual blocks per layer | 2 |
| Channels in convolutions | 64 |
| Self-attention layers at resolution | 8 / 16 |

Table 3: Embedding table hyperparameters.

| Hyperparameter | Value |
| --- | --- |
| Vocabulary size (N) | 512 |
| Tokens per frame (K) | 16 |
| Token embedding dimension (d) | 512 |

Note that during experience collection in the real environment, frames still go through the autoencoder to keep the input distribution of the policy unchanged. See Algorithm 1 for details.

## A.2  TRANSFORMER

Our autoregressive Transformer is based on the implementation of minGPT (Karpathy, 2020). It takes as input a sequence of $L(K + 1)$ tokens and embeds it into a $L(K + 1) \times D$ tensor using an $A \times D$ embedding table for actions, and a $N \times D$ embedding table for frames tokens. This tensor is forwarded through $M$ Transformer blocks. We use GPT2-like blocks (Radford et al., 2019), i.e. each block consists of a self-attention module with layer normalization of the input, wrapped with a residual connection, followed by a per-position multi-layer perceptron with layer normalization of the input, wrapped with another residual connection.

Table 4: Transformer hyperparameters

| Hyperparameter | Value |
|---|---|
| Timesteps (L) | 20 |
| Embedding dimension (D) | 256 |
| Layers (M) | 10 |
| Attention heads | 4 |
| Weight decay | 0.01 |
| Embedding dropout | 0.1 |
| Attention dropout | 0.1 |
| Residual dropout | 0.1 |

## A.3 ACTOR-CRITIC

The weights of the actor and critic are shared except for the last layer. The actor-critic takes as input a $64 \times 64 \times 3$ frame, and forwards it through a convolutional block followed by an LSTM cell (Mnih et al., 2016; Hochreiter & Schmidhuber, 1997; Gers et al., 2000). The convolutional block consists of the same layer repeated four times: a 3x3 convolution with stride 1 and padding 1, a ReLU activation, and 2x2 max-pooling with stride 2. The dimension of the LSTM hidden state is 512. Before starting the imagination procedure from a given frame, we burn-in (Kapturowski et al., 2019) the 20 previous frames to initialize the hidden state.

Table 5: Training loop & Shared hyperparameters

| Hyperparameter | Value |
|---|---|
| Epochs | 600 |
| # Collection epochs | 500 |
| Environment steps per epoch | 200 |
| Collection epsilon-greedy | 0.01 |
| Eval sampling temperature | 0.5 |
| Start autoencoder after epochs | 5 |
| Start transformer after epochs | 25 |
| Start actor-critic after epochs | 50 |
| Autoencoder batch size | 256 |
| Transformer batch size | 64 |
| Actor-critic batch size | 64 |
| Training steps per epoch | 200 |
| Learning rate | 1e-4 |
| Optimizer | Adam |
| Adam $\beta_1$ | 0.9 |
| Adam $\beta_2$ | 0.999 |
| Max gradient norm | 10.0 |

# B  ACTOR-CRITIC LEARNING OBJECTIVES

We follow Dreamer (Hafner et al., 2020; 2021) in using the generic $\lambda$-return, that balances bias and variance, as the regression target for the value network. Given an imagined trajectory $(\hat{x}_0, a_0, \hat{r}_0, \hat{d}_0, \ldots, \hat{x}_{H-1}, a_{H-1}, \hat{r}_{H-1}, \hat{d}_{H-1}, \hat{x}_H)$, the $\lambda$-return can be defined recursively as follows:

$$\Lambda_t = \begin{cases} \hat{r}_t + \gamma(1 - \hat{d}_t)\Big[(1 - \lambda)V(\hat{x}_{t+1}) + \lambda\Lambda_{t+1}\Big] & \text{if} \quad t < H \\ V(\hat{x}_H) & \text{if} \quad t = H \end{cases} \tag{4}$$

The value network $V$ is trained to minimize $\mathcal{L}_V$, the expected squared difference with $\lambda$-returns over imagined trajectories.

$$\mathcal{L}_V = \mathbb{E}_\pi\Big[\sum_{t=0}^{H-1}\big(V(\hat{x}_t) - \text{sg}(\Lambda_t)\big)^2\Big] \tag{5}$$

Here, $\text{sg}(\cdot)$ denotes the gradient stopping operation, meaning that the target is a constant in the gradient-based optimization, as classically established in the literature (Mnih et al., 2015; Hessel et al., 2018; Hafner et al., 2020).

As large amounts of trajectories are generated in the imagination MDP, we can use a straightforward reinforcement learning objective for the policy, such as REINFORCE (Sutton & Barto, 2018). To reduce the variance of REINFORCE gradients, we use the value $V(\hat{x}_t)$ as a baseline (Sutton & Barto, 2018). We also add a weighted entropy maximization objective to maintain a sufficient exploration. The actor is trained to minimize the following REINFORCE objective over imagined trajectories:

$$\mathcal{L}_\pi = -\mathbb{E}_\pi\Big[\sum_{t=0}^{H-1}\log(\pi(a_t|\hat{x}_{\leq t}))\,\text{sg}(\Lambda_t - V(\hat{x}_t)) + \eta\,\mathcal{H}(\pi(a_t|\hat{x}_{\leq t}))\Big] \tag{6}$$

Table 6: RL training hyperparameters

| Hyperparameter | Value |
|---|---|
| Imagination horizon (H) | 20 |
| $\gamma$ | 0.995 |
| $\lambda$ | 0.95 |
| $\eta$ | 0.001 |

## C  OPTIMALITY GAP

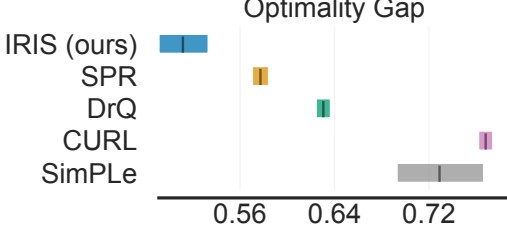

Figure 8: Optimality gap, lower is better. The amount by which the algorithm fails to reach a human-level score (Agarwal et al., 2021).

## D  IRIS ALGORITHM

---
**Algorithm 1:** IRIS

---
**Procedure** `training_loop()`:
  **for** *epochs* **do**
    `collect_experience(`*steps_collect*`)`
    **for** *steps_world_model* **do**
      `update_world_model()`
    **for** *steps_behavior* **do**
      `update_behavior()`

**Procedure** `collect_experience(`$n$`)`:
  $x_0 \leftarrow$ `env.reset()`
  **for** $t = 0$ **to** $n - 1$ **do**
    $\hat{x}_t \leftarrow D(E(x_t))$  `//` forward frame through discrete autoencoder
    Sample $a_t \sim \pi(a_t | \hat{x}_t)$
    $x_{t+1}, r_t, d_t \leftarrow$ `env.step(`$a_t$`)`
    **if** $d_t = 1$ **then**
      $x_{t+1} \leftarrow$ `env.reset()`
  $\mathcal{D} \leftarrow \mathcal{D} \cup \{x_t, a_t, r_t, d_t\}_{t=0}^{n-1}$

**Procedure** `update_world_model()`:
  Sample $\{x_t, a_t, r_t, d_t\}_{t=\tau}^{\tau+L-1} \sim \mathcal{D}$
  Compute $z_t := E(x_t)$ and $\hat{x}_t := D(z_t)$ for $t = \tau, \ldots, \tau + L - 1$
  Update $E$ and $D$
  Compute $p_G(\hat{z}_{t+1}, \hat{r}_t, \hat{d}_t \mid z_\tau, a_\tau, \ldots, z_t, a_t)$ for $t = \tau, \ldots, \tau + L - 1$
  Update $G$

**Procedure** `update_behavior()`:
  Sample $x_0 \sim \mathcal{D}$
  $z_0 \leftarrow E(x_0)$
  $\hat{x}_0 \leftarrow D(z_0)$
  **for** $t = 0$ **to** $H - 1$ **do**
    Sample $a_t \sim \pi(a_t | \hat{x}_t)$
    Sample $\hat{z}_{t+1}, \hat{r}_t, \hat{d}_t \sim p_G(\hat{z}_{t+1}, \hat{r}_t, \hat{d}_t \mid z_0, a_0, \ldots, \hat{z}_t, a_t)$
    $\hat{x}_{t+1} \leftarrow D(\hat{z}_{t+1})$
  Compute $V(\hat{x}_t)$ for $t = 0, \ldots, H$
  Update $\pi$ and $V$

---

# E    AUTOENCONDING FRAMES WITH VARYING AMOUNTS OF TOKENS

The sequence length of the Transformer is determined by the number of tokens used to encode a single frame and the number of timesteps in memory. Increasing the number of tokens per frame results in better reconstructions, although it requires more compute and memory.

This tradeoff is particularly important in visually challenging games with a high number of possible configurations, where the discrete autoencoder struggles to properly encode frames with only 16 tokens. For instance, Figure 9 shows that, when increasing the number of tokens per frame to 64 in *Alien*, the discrete autoencoder correctly reconstructs the player, its enemies, and rewards.

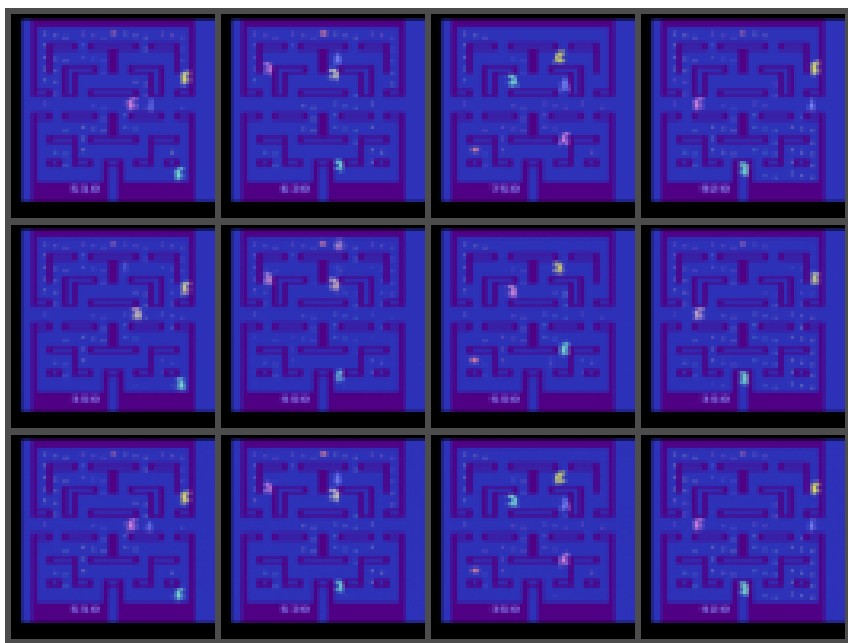

Figure 9: Tradeoff between the number of tokens per frame and reconstructions quality in *Alien*. Each column displays a $64 \times 64$ frame from the real environment (top), its reconstruction with a discrete encoding of 16 tokens (center), and its reconstruction with a discrete encoding of 64 tokens (bottom). In *Alien*, the player is the dark blue character, and the enemies are the large colored sprites. With 16 tokens per frame, the autoencoder often erases the player, switches colors, and misplaces rewards. When increasing the amount of tokens, it properly reconstructs the frame.

Table 7 displays the final performance of IRIS trained with 64 tokens per frame in three games. Interestingly, even though the world model is more accurate, the performance in Alien only increases marginally (+36%). This observation suggests that Alien poses a hard reinforcement learning problem, as evidenced by the low performance of other baselines in that game. On the contrary, IRIS greatly benefits from having more tokens per frame for Asterix (+121%) and BankHeist (+432%).

Table 7: Returns on Alien, Asterix, and BankHeist with 64 tokens per frame instead of 16.

| Game | Random | Human | SimPLe | CURL | DrQ | SPR | IRIS (16 tokens) | IRIS (64 tokens) |
|------|--------|-------|--------|------|-----|-----|------------------|------------------|
| Alien | 227.8 | 7127.7 | 616.9 | 711.0 | **865.2** | 841.9 | 420.0 | 570.0 |
| Asterix | 210.0 | 8503.3 | 1128.3 | 567.2 | 763.6 | 962.5 | 853.6 | **1890.4** |
| BankHeist | 14.2 | 753.1 | 34.2 | 65.3 | 232.9 | **345.4** | 53.1 | 282.5 |

## F  BEYOND THE SAMPLE-EFFICIENT SETTING

IRIS can be scaled up by increasing the number of tokens used to encode frames, adding capacity to the model, taking more optimization steps per environment steps, or using more data. In this experiment, we investigate data scaling properties by increasing the number of environment steps from 100k to 10M. However, to maintain a training time within our computational resources, we lower the ratio of optimization steps per environment steps from 1:1 to 1:50. As a consequence, the results of this experiment at 100k frames would be worse than those reported in the paper.

Table 8: Increasing the number of environment steps from 100k to 10M.

| Game | Random | Human | IRIS (100k) | IRIS (10M) |
|---|---|---|---|---|
| Alien | 227.8 | 7127.7 | 420.0 | 1003.1 |
| Amidar | 5.8 | 1719.5 | 143.0 | 213.4 |
| Assault | 222.4 | 742.0 | 1524.4 | 9355.6 |
| Asterix | 210.0 | 8503.3 | 853.6 | 6861.0 |
| BankHeist | 14.2 | 753.1 | 53.1 | 921.6 |
| BattleZone | 2360.0 | 37187.5 | 13074.0 | 34562.5 |
| Boxing | 0.1 | 12.1 | 70.1 | 98.0 |
| Breakout | 1.7 | 30.5 | 83.7 | 493.9 |
| ChopperCommand | 811.0 | 7387.8 | 1565.0 | 9814.0 |
| CrazyClimber | 10780.5 | 35829.4 | 59324.2 | 111068.8 |
| DemonAttack | 152.1 | 1971.0 | 2034.4 | 96218.6 |
| Freeway | 0.0 | 29.6 | 31.1 | 34.0 |
| Frostbite | 65.2 | 4334.7 | 259.1 | 290.3 |
| Gopher | 257.6 | 2412.5 | 2236.1 | 97370.6 |
| Hero | 1027.0 | 30826.4 | 7037.4 | 19212.0 |
| Jamesbond | 29.0 | 302.8 | 462.7 | 5534.4 |
| Kangaroo | 52.0 | 3035.0 | 838.2 | 1793.8 |
| Krull | 1598.0 | 2665.5 | 6616.4 | 7344.0 |
| KungFuMaster | 258.5 | 22736.3 | 21759.8 | 39643.8 |
| MsPacman | 307.3 | 6951.6 | 999.1 | 1233.0 |
| Pong | -20.7 | 14.6 | 14.6 | 21.0 |
| PrivateEye | 24.9 | 69571.3 | 100.0 | 100.0 |
| Qbert | 163.9 | 13455.0 | 745.7 | 4012.1 |
| RoadRunner | 11.5 | 7845.0 | 9614.6 | 30609.4 |
| Seaquest | 68.4 | 42054.7 | 661.3 | 1815.0 |
| UpNDown | 533.4 | 11693.2 | 3546.2 | 114690.1 |
| #Superhuman (↑) | 0 | N/A | 10 | 15 |
| Mean (↑) | 0.000 | 1.000 | 1.046 | 7.488 |
| Median (↑) | 0.000 | 1.000 | 0.289 | 1.207 |
| IQM (↑) | 0.000 | 1.000 | 0.501 | 2.239 |
| Optimality Gap (↓) | 1.000 | 0.000 | 0.512 | 0.282 |

Table 8 illustrates that increasing the number of environment steps from 100k to 10M drastically improves performance for most games, providing evidence that IRIS could be scaled up beyond the sample-efficient regime. On some games, more data only yields marginal improvements, most likely due to hard exploration problems or visually challenging domains that would benefit from a higher number of tokens to encode frames (Appendix E).

## G    COMPUTATIONAL RESOURCES

For each Atari environment, we repeatedly trained IRIS with 5 different random seeds. We ran our experiments with 8 Nvidia A100 40GB GPUs. With two Atari environments running on the same GPU, training takes around 7 days, resulting in an average of 3.5 days per environment.

SimPLe (Kaiser et al., 2020), the only baseline that involves learning in imagination, trains for 3 weeks with a P100 GPU on a single environment. As for SPR (Schwarzer et al., 2021), the strongest baseline without lookahead search, it trains notably fast in 4.6 hours with a P100 GPU.

Regarding baselines with lookahead search, MuZero (Schrittwieser et al., 2020) originally used 40 TPUs for 12 hours to train in a single Atari environment. Ye et al. (2021) train both EfficientZero and their reimplementation of MuZero in 7 hours with 4 RTX 3090 GPUs. EfficientZero's implementation relies on a distributed infrastructure with CPU and GPU threads running in parallel, and a C++/Cython implementation of MCTS. By contrast, IRIS and the baselines without lookahead search rely on straightforward single GPU / single CPU implementations.

## H    EXPLORATION IN FREEWAY

The reward function in Freeway is sparse since the agent is only rewarded when it completely crosses the road. In addition, bumping into cars will drag it down, preventing it from smoothly ascending the highway. This poses an exploration problem for newly initialized agents because a random policy will almost surely never obtain a non-zero reward with a 100k frames budget.

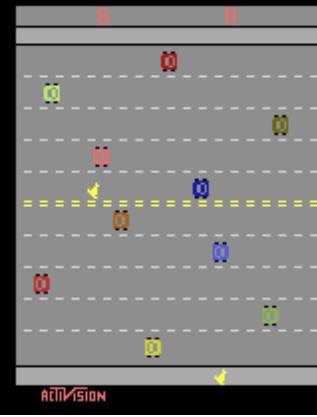

Figure 10: A game of *Freeway*. Cars will bump the player down, making it very unlikely to cross the road and be rewarded for random policies.

The solution to this problem is actually straightforward and simply requires stretches of time when the UP action is oversampled. Most Atari 100k baselines fix the issue with epsilon-greedy schedules and argmax action selection, where at some point the network configuration will be such that the UP action is heavily favored. In this work, we opted for the simpler strategy of having a fixed epsilon-greedy parameter and sampling from the policy. However, we lowered the sampling temperature from 1 to 0.01 for Freeway, in order to avoid random walks that would not be conducive to learning in the early stages of training. As a consequence, once it received its first few rewards through exploration, IRIS was able to internalize the sparse reward function in its world model.

