# OpenReview forum: "Transformers are Sample-Efficient World Models"
_ICLR.cc/2023/Conference — ICLR 2023 notable top 5%_

### Official Review · Reviewer_fmDi · 2022-10-19

**Confidence:** 4
**Correctness:** 3
**Technical Novelty And Significance:** 4
**Empirical Novelty And Significance:** 4
**Recommendation:** 8

**Clarity, Quality, Novelty And Reproducibility:**

The submission is clear, of high quality, and novel. The submission includes code.

**Strength And Weaknesses:**

### Strengths

- Loosely speaking, existing successful MBRL algorithms are modifications of Dreamer or MuZero. Showing that another framework for MBRL can be successful is a valuable contribution.
- The submission largely follows the guidelines for benchmarking laid out by the statistical precipice paper.

### Weaknesses

- At least in my opinion, Atari 100k is not particularly well benchmarked. I would say that the only "good" algorithm that has been benchmarked by experimenters incentivized to tune the algorithm to maximize performance is EfficientZero. Thus, the extent to which IRIS is good is a bit unclear to me. It would be interesting to see how IRIS performs on the standard Atari benchmark (or, inversely, how something like Dreamer compares to IRIS on Atari 100k).
- The statistical precipice paper suggests that results on Atari 100k are reliable (under appropriate metrics) with as few as 10 runs; the submission only uses 5 runs. My gut feeling is that the margin of improvement under various metrics seems substantial enough that it would probably continue to hold under a more reliable number of runs, but it would be good to actually show this.

---

Addendum:
I also concur with reviewer t9Kq: 1) the submission would benefit from additional attention to related work (such as [1],[2],[3]) and 2) additional ablations.

### Comments on decision-time planning:

The submission argues: "Moreover, IRIS could be combined with MCTS, both in imagination and in the real environment. Therefore, methods involving lookahead search should not be seen as direct competitors but rather as potential extensions to learning-only methods." In principle, this is of course true. However, as a matter of practice, I am not sure it is as clear. I am not aware of any examples of a non-MuZero-like architecture successfully utilizing decision-time planning. It is plausible to me that algorithms like Dreamer and IRIS, which perform well in a background planning regime, may not enjoy much benefit from decision-time MCTS.

### Comments on superhuman performance:

> Most notably, human experts were surpassed by deep RL algorithms in a multitude of arcade (Mnih et al., 2015; Schrittwieser et al., 2020; Hafner et al., 2021), real-time strategy (Vinyals et al., 2019; Berner et al., 2019), board (Silver et al., 2016; 2018; Schrittwieser et al., 2020) and imperfect information (Schmid et al., 2021; Brown et al., 2020a) games.

I find this sentence is misleading. Deep RL algorithms have not surpassed human experts in most Atari games, as is clearly evidenced by the human world record metric. They have also not surpassed human experts in StarCraft (AlphaStar is only grandmaster level) or DOTA (OpenAI5 played a restricted version of the game and was found to be reliably exploitable by humans).

**Summary Of The Paper:**

The submission benchmarks a transformer-based approach to background planning in the Atari 100k benchmark. The submission finds that its approach outperforms existing methods that have benchmarked on Atari 100k, excluding those that use decision-time planning.

**Summary Of The Review:**

The submission shows a qualitatively novel approach to MBRL can, to some extent, be successful. My main gripe is that the submission shows these results in a domain that is, in my opinion, not well benchmarked; thus, IRIS's strength as a MBRL algorithm is a bit unclear to me. Nevertheless, I think the evidence presented in the submission is sufficiently strong to merit acceptance as is.

---

> ### Author Response · Authors · 2022-11-18
> **Response to Reviewer fmDi**
>
> We thank reviewer fmDi for their thorough and insightful review.
>
> ---
>
> >It would be interesting to see how IRIS performs on the standard Atari benchmark (or, inversely, how something like Dreamer compares to IRIS on Atari 100k).
>
> We agree that the benchmark would greatly benefit from the inclusion of DreamerV2. It would also be interesting to scale up IRIS beyond the sample-efficient setting, and we plan on doing so in future work.
>
> ---
>
> >The statistical precipice paper suggests that results on Atari 100k are reliable (under appropriate metrics) with as few as 10 runs; the submission only uses 5 runs. My gut feeling is that the margin of improvement under various metrics seems substantial enough that it would probably continue to hold under a more reliable number of runs, but it would be good to actually show this.
>
> Interestingly, stratified bootstrap confidence intervals work remarkably well even with 5 seeds. From Section A.2 of the statistical precipice paper: “on Atari 100k, for achieving true coverage close to 95%, such CIs [bootstrap CIs from Colas et al.] require at least 20-30 runs per task as opposed to 5-10 runs for stratified bootstrap CIs for aggregate metrics like median, mean and IQM”. In addition, Figure A.19 “coverage vs number of runs” shows that, with 5 runs, more than 90% of stratified bootstrap CIs cover robust estimates of the mean, median, and IQM. In any case, we agree that 10 runs would be better.
>
> ---
>
> >Addendum: [...] 1) the submission would benefit from additional attention to related work (such as [1],[2],[3]) and 2) additional ablations.
>
> We discussed the related work mentioned and ran additional experiments. More details can be found in our answers to reviewers [CZra](https://openreview.net/forum?id=vhFu1Acb0xb&noteId=PX3ZJq8F8A) and [t9Kq](https://openreview.net/forum?id=vhFu1Acb0xb&noteId=1nqVA_RcXTG).
>
> ---
>
> >Comments on decision-time planning [...] I am not aware of any examples of a non-MuZero-like architecture successfully utilizing decision-time planning. It is plausible to me that algorithms like Dreamer and IRIS, which perform well in a background planning regime, may not enjoy much benefit from decision-time MCTS.
>
> We are very enthusiastic about combining IRIS with decision-time planning but we completely agree with that comment since we did not run any experiments yet. Thanks for this judicious remark. We have updated Section 3.1 and the conclusion accordingly.
>
> ---
>
> >Comments on superhuman performance [...] Deep RL algorithms have not surpassed human experts in most Atari games, as is clearly evidenced by the human world record metric. They have also not surpassed human experts in StarCraft (AlphaStar is only grandmaster level) or DOTA (OpenAI5 played a restricted version of the game and was found to be reliably exploitable by humans).
>
> Thanks for pointing that out. We have fixed this inaccuracy in the revised version.

---

> > ### Comment · Reviewer_fmDi · 2022-11-19
> > **Response**
> >
> > Thanks for these comments! I find the authors responses very reasonable -- congrats on the great paper! :)

---

### Official Review · Reviewer_t9Kq · 2022-10-23

**Confidence:** 4
**Correctness:** 3
**Technical Novelty And Significance:** 3
**Empirical Novelty And Significance:** 3
**Recommendation:** 8

**Clarity, Quality, Novelty And Reproducibility:**

The article is very well read and the information given is very clear. My main concern could found above.

**Strength And Weaknesses:**


[Pros]
* The paper is well-writen and clear to read. the effieciecency and effacacy are good on the atari100k benchmark.
* The code is clear and easy to follow.

[Cons]
* Noveltiy is insufficient (Some disucussion to [1][2][3] would be appreciated)
* Increasing the number of ablations and analyses to improve comprehension of the contributions of various designed parts.(e.g. different tokenizer, backbone)

[1] TRANSDREAMER: REINFORCEMENT LEARNING WITH TRANSFORMER WORLD MODELS https://arxiv.org/pdf/2202.09481.pdf
[2] Reinforcement Learning with Action-Free Pre-Training from Videos, https://arxiv.org/pdf/2203.13880.pdf
[3] Online Decision Transformer, https://arxiv.org/pdf/2202.05607.pdf


*Some typos: sec2.2 blue arrow-> purple? arrow

**Summary Of The Paper:**

This paper proposes a miniGPT-like transformer architecture for learning the world model of RL agents in POMDP environments. This World Model's training data is derived from the present policy model's interplay with the real world. The input and output images are then represented in the VQGAN-style and utilized to train the global model. Finally, the policy function of the RL Agent is trained using the fictitious world model data. The authors demonstrate the efficacy of this method on the Atari 100k benchmark, which achieves a mean human
normalized score of 1.046, and outperforms humans on 10 out of 26 games with two-hours training data.

**Summary Of The Review:**

Overall, I think this article is worth reading and being published, as it explores the possibilities of using Transformer to learn World Model. Although I am not a Big Fan of the World model, which breaks down reinforcement learning into learning a simulater for the environment and then learning it, I still endorse the paper's conclusion that Transformer-based architectures (Vit, Swin, etc.) can encode image inputs into a series that can be easily understood and learned tokens.

However, I still hope the author can give more in-depth understanding, such as how the encoding of the world-model itself accounts for this task, and whether the pure necessity of the world-model itself is sufficient (e.g., the same backbone structure of the A2C agent can be used, etc.)
------------------------------------------------------------------------------------------------------
I have raise my score to 8, while there are still many details I'd like to know about this paper, overall, I'm inclined to think it deserves to be published.

---

> ### Author Response · Authors · 2022-11-18
> **Response to Reviewer t9Kq**
>
> We thank Reviewer t9Kq for their positive feedback. We are glad that they took the time to dive into the codebase.
>
> ---
>
> >Noveltiy is insufficient (Some disucussion to [1][2][3] would be appreciated)
>
> We thank Reviewer t9Kq for these two additional references ([2] and [3]). We have included them in the revised version. As discussed below, we do not think they limit the novelty of our contribution.
>
> **TransDreamer** [1]
>
> As mentioned in the original related work section, TransDreamer is a modified DreamerV2 where a Transformer replaces the GRU in the Recurrent State Space Model, resulting in an architecture that differs in many regards with our method (KL balancing, no autoregression over latent tokens, joint training instead of two stage training, etc).
>
> The paper reports results on four Atari games, three of which are featured in Atari 100k: Boxing, Freeway and Pong. IRIS outperforms TransDreamer on these three games, achieving comparable returns with respectively 175x, 120x, and 400x less environment steps.
>
> More thorough experiments would be needed to understand what structural differences explain this performance gap. To the best of our knowledge, the authors do not provide any code for further comparisons.
>
> **Reinforcement Learning with Action-Free Pre-Training from Videos** [2]
>
> This paper extends DreamerV2 to the scenario where an offline video dataset is available. In a first stage, the authors pretrain a recurrent video prediction model on this dataset. In a second stage, the images are fed to the video prediction model, whose representations become the input of a standard DreamerV2 agent.
>
> It is not clear to us why the novelty of our method is diminished by this paper since their contribution is to enhance DreamerV2 when a video dataset is available, which does not feature a Transformer-based world model.
>
> **Online Decision Transformer** [3]
>
> In this work, the authors propose a unified framework for offline pretraining and online finetuning of Decision Transformers.
>
> As discussed in the Related Work section, Decision Transformers are policies and not world models. Therefore, we don’t think they limit the novelty of our contribution given that we focus on a different problem, namely learning in imagination with Transformer-based world models.
>
> ---
>
> >Increasing the number of ablations and analyses to improve comprehension of the contributions of various designed parts.(e.g. different tokenizer, backbone)
>
> We agree with Reviewer t9Kq that ablations are important to deepen one’s understanding of the method. Therefore, following the reviewers’ suggestions, we ran two additional experiments:
>
> - We investigated the effect of the number of tokens per frame on final performance. We originally showed that for visually challenging games, having a small number of tokens resulted in poor reconstructions and world modeling (see the original Appendix E). To further our understanding, we trained IRIS with more tokens per frame (64 instead of 16) in 3 games. We observe that for Asterix and BankHeist performance greatly increases (+121% and +432% respectively), which indicates that having a small number of tokens was a bottleneck. For Alien, even though the world model is more accurate, performance only increases marginally (+36%). This suggests that Alien poses a hard RL problem, as evidenced by the low performance of other baselines in that game. This experiment is found in Appendix E of the revised paper.
>
> - We made sure that IRIS keeps improving as we increase the number of training frames from 100k to 6M. IRIS performs much better at 6M frames than at 100k frames (x5 on Breakout, x41 on DemonAttack, x32 on Gopher). Please see the [answer to Reviewer CZra](https://openreview.net/forum?id=vhFu1Acb0xb&noteId=PX3ZJq8F8A) for more details. We plan on extending this experiment to the full suite of games and to include it in the final version.
>
> ---
>
> >I still hope the author can give more in-depth understanding, [...] whether the pure necessity of the world-model itself is sufficient (e.g., the same backbone structure of the A2C agent can be used, etc.)
>
> We agree that it is important to make sure that having a world model does not only yield a marginal performance gain compared to direct reinforcement learning in the environment. Kaiser et al. [4] showed that standard model-free agents such as Rainbow and PPO struggled in Atari 100k, thus justifying the need for a world model like SimPLe in this regime. From what we observed during the development of IRIS, the actor-critic alone, without a world model to learn from, did not discover any rewarding behaviors with only 100k environmental interactions.
>
> ---
>
> >Some typos: sec2.2 blue arrow-> purple? arrow
>
> Thanks for the comment. The current text is correct since the blue arrow depicts the Transformer and section 2.2 is about the Transformer.
>
> We hope to have addressed Reviewer t9Kq’s concerns regarding the novelty of our method and ablations.

---

> > ### Author Response · Authors · 2022-11-18
> > **References**
> >
> > [1] Chang Chen, Yi-Fu Wu, Jaesik Yoon, and Sungjin Ahn. Transdreamer: Reinforcement learning with transformer world models. arXiv preprint arXiv:2202.09481, 2022.
> >
> > [2] Younggyo Seo, Kimin Lee, Stephen L James, and Pieter Abbeel. Reinforcement learning with action- free pre-training from videos. In International Conference on Machine Learning, pp. 19561–19579. PMLR, 2022.
> >
> > [3] Qinqing Zheng, Amy Zhang, and Aditya Grover. Online decision transformer. In International Conference on Machine Learning, pp. 27042–27059. PMLR, 2022.
> >
> > [4] Łukasz Kaiser, Mohammad Babaeizadeh, Piotr Miłos, Blazej Osinski, et al. Model based reinforcement learning for atari. In International Conference on Learning Representations, 2020.

---

### Official Review · Reviewer_CZra · 2022-10-26

**Confidence:** 3
**Correctness:** 3
**Technical Novelty And Significance:** 3
**Empirical Novelty And Significance:** 3
**Recommendation:** 8

**Clarity, Quality, Novelty And Reproducibility:**

The paper is well-written, and the models and results are clearly presented. The idea of using Transformer as world models is interesting and novel to me.

**Strength And Weaknesses:**

**Strength**:

1. Using Transformers as world models in model-based RL seems novel.
2. The results are promising.

**Weaknesses**:

1. Although it is emphasized in the paper that sample efficiency should be a main consideration for model-based RL methods, I think training for a longer time and show that the proposed methods can achieve good results on most games would also be worth considering to verify the proposed models.
2. The objectives of training the autoencoder and Transformer is missing in the paper and in the appendix (only mentioned by a few sentences). It would be better to also include those settings in the appendix.

**Summary Of The Paper:**

This work proposes IRIS, which uses a world model to train agents. The world model consists of a discrete autoencoder and an autoregressive Transformer.

The discrete autoencoder $(E, D)$ consists of an encoder $E$, which converts an input image to tokens, and a decoder, which turns tokens back to an image. Given previous tokens and actions, the autoregressive Transformer then models (1) the transitions to next token; (2) reward; (3) termination of the episode.

The world model is then used to train an actor-critic method using standard reinforcement learning (RL) objectives.

The authors conducted experiments on Atari 100k benchmark. After 100k actions (about 2 hours of human gameplay experience), the proposed IRIS method can outperform several other baselines including SimPLe, CURL, and DrQ, and it outperforms human players on 10 out of 26 games.

The authors also show results of the good prediction ability of Transformer world models in terms of accurately predicting next status of Atari game images, and rewards. Some drawbacks of the world models, such as inability of predicting some games with multiple layers are also discussed.

**Summary Of The Review:**

Overall, this work studies an interesting idea of using Transformer as world models in model-based RL, and the promising experimental results support the proposed methods well.

**Update after rebuttal**:

I would like to thank the authors for the feedback, which addressed my main concerns. I increased my score accordingly.

---

> ### Author Response · Authors · 2022-11-18
> **Response to Reviewer CZra**
>
> We thank Reviewer CZra for their detailed review.
>
> ---
>
> >Although it is emphasized in the paper that sample efficiency should be a main consideration for model-based RL methods, I think training for a longer time and show that the proposed methods can achieve good results on most games would also be worth considering to verify the proposed models.
>
> We agree that training sample-efficient methods with a bigger frame budget is important to make sure that they do not stop improving beyond the original constraint.
>
> In that spirit, we ran an additional experiment with a budget of 6M frames on Breakout, DemonAttack and Gopher. We kept the same hyperparameters and only decreased the relative number of training steps to avoid multiplying the training time by a factor of 60 (6M/100K). As a consequence, the results of these runs at 100k frames are worse than those reported in the paper.
>
> Here are the returns and human-normalized scores obtained:
>
> |             | 100k (paper)      | 6M (here)          |
> |-------------|-------------------|--------------------|
> | Breakout    | 83.7 (2.85 HNS)   | 406.1 (14.04 HNS)  |
> | DemonAttack | 2034.4 (1.03 HNS) | 77273.5 (42.4 HNS) |
> | Gopher      | 2236.1 (0.92 HNS) | 64251.6 (29.7 HNS) |
>
> IRIS performs much better at 6M frames than at 100k frames (x5 on Breakout, x41 on DemonAttack, x32 on Gopher)  even without scaling the models, which would probably benefit from more data.
>
> We thank Reviewer CZra for this suggestion. We will run this experiment on the full suite of games and include the results in the Appendix for the final version.
>
> ---
>
> >The objectives of training the autoencoder and Transformer is missing in the paper and in the appendix (only mentioned by a few sentences). It would be better to also include those settings in the appendix.
>
> Thanks for pointing that out. We have updated Appendix A.1 to include the training objective of the discrete autoencoder. Regarding the Transformer, we already specify the distributions being modelled in section 2.2 and the learning objective is a standard cross-entropy between these distributions and the labels (i.e. next state token, reward, and episode end).
>
> ---
>
> We hope to have addressed Reviewer CZra’s comments regarding the ability of the method to scale with more data and the training objectives.

---

### Official Review · Reviewer_Knzj · 2022-10-28

**Confidence:** 5
**Correctness:** 4
**Technical Novelty And Significance:** 4
**Empirical Novelty And Significance:** 2
**Recommendation:** 8

**Clarity, Quality, Novelty And Reproducibility:**

Creative repurposing of existing components. The method seems to be easy to use and re-implement.

**Strength And Weaknesses:**

Strengths:

1. Simple and easy-to-understand learning algorithm.
2. Well-thought-through reuse of standard modeling and algorithmic components.
3. Outstanding results on Atari 100K benchmark, in particular, bearing in mind the limited compute and expressivity of models.

Weaknesses:

1. Models and agents are trained per game. I am curious what would happen if one tries to train a multi-game world model and agent in the spirit of Gato https://arxiv.org/abs/2205.06175 or Multi-game decision transformers https://arxiv.org/abs/2205.15241.
2. Nit: the paper may benefit from using more modest language. I mean, in particular, the “drastically different architecture” claim or not very elegant comparisons to look-ahead algorithms.

A bug that may turn into a feature:

The method needs “more tokens” for games that require the detailed representation of images (as authors put it: “Another kind of games difficult to simulate are mazes with moving enemies, such as MsPacman, BankHeist, and Alien”). This characteristic seems to be a limitation but may lead to an interesting long-context benchmark. Some environments may require more frames and details, leading to a new dataset that can be used to benchmark long-context transformer models.

**Summary Of The Paper:**

Authors train very impressive agents for the 100K Atari benchmark. The method consists in representing Atari frames with discrete tokens and training a transformer-based world model and a game agent. The transformer model is trained in a supervised way. The agent is trained entirely using frames from the world model.

**Summary Of The Review:**

Authors train very impressive agents for the 100K Atari benchmark.  Benchmark results are excellent but algorithmic, and modeling novelty is limited.

---

> ### Author Response · Authors · 2022-11-18
> **Response to Reviewer Knzj**
>
> We thank Reviewer Knzj for their insightful and positive review.
>
> ---
>
> >Models and agents are trained per game. I am curious what would happen if one tries to train a multi-game world model and agent in the spirit of Gato or Multi-game decision transformers.
>
> This is a very interesting suggestion. The current networks are most likely under-parameterized for the multi-game setup, and should probably be scaled up. It would indeed be fascinating to have a single world model capable of handling a variety of games with various dynamics, and to investigate transfer effects between games.
>
> ---
>
> >Nit: the paper may benefit from using more modest language. I mean, in particular, the “drastically different architecture” claim or not very elegant comparisons to look-ahead algorithms.
>
> Thanks for pointing that out. We have updated the paper to reflect these comments, in particular section 3.1 and the introduction.
>
> ---
>
> >The method needs “more tokens” for games that require the detailed representation of images (as authors put it: “Another kind of games difficult to simulate are mazes with moving enemies, such as MsPacman, BankHeist, and Alien”). This characteristic seems to be a limitation but may lead to an interesting long-context benchmark. Some environments may require more frames and details, leading to a new dataset that can be used to benchmark long-context transformer models.
>
> Indeed, the number of tokens in the sequence is directed related to the “complexity” of the frames to encode. Hence, we may have to deal with a long context because of time dependencies or because a fine-grained visual understanding is necessary to model dynamics. We agree that a new benchmark would be useful to evaluate long-context architectures on this class of problems.

---

### Author Response · Authors · 2022-11-18
**General response**

We thank the reviewers for their thorough reviews. Based on the reviewers’ comments, we have updated the paper in several ways:

- We added an ablation to measure the impact of the number of tokens on final performance (see [response to Reviewer t9Kq](https://openreview.net/forum?id=vhFu1Acb0xb&noteId=1nqVA_RcXTG) and the updated Appendix E).

- We ran an experiment to ensure that IRIS keeps improving beyond the original 100k constraint (see [response to Reviewer CZra](https://openreview.net/forum?id=vhFu1Acb0xb&noteId=PX3ZJq8F8A), results on the full suite of games will be added to the appendix).

- We discussed the related work suggested by Reviewer t9Kq and updated the Related Work section accordingly.

- We reflected comments made by Reviewers Knzj and fmDi in the introduction, section 3.1, and the conclusion.

We are thankful for this overall very positive feedback and we look forward to the second stage of the discussion period!

---

### Decision · Program_Chairs · 2023-01-20

**Decision:**

Accept: notable-top-5%

**Justification For Why Not Higher Score:**

N/A

**Justification For Why Not Lower Score:**

Data efficient RL is an interesting and important problem and this paper discusses a very simple but effective solution for visual domains. I think people will likely build on top of it and so it deserves to have an oral.

**Metareview: Summary, Strengths And Weaknesses:**

This paper studies the problem of learning to behave purely from a transformer based transition model trained from experiences collected in the Atari domain. The visual frames are tokenizes (VQ-GAN) before feeding into the transformer. The transformer predicts the future world state for a policy to learn goal-directed behaviors. Interestingly, the goal directed policy is only trained from experiences sampled from the transformer + VQ-GAN decoder (not the original data). I would have expected that learning on top of predicted pixels would accumulate large amount of noise due to drifting but it seems to be good enough to play reasonably well in Atari.

The approach is extremely easy to understand and to implement. The results and experiments are on solid foundations. The authors demonstrate this method on the Atari 100k benchmark and achieves a a mean human normalized score of 1.046, and outperforms humans on 10 out of 26 games with two-hours training data. The baselines are systematic and it does not seem like anything major is missing with respect to that.

The clarity of the paper is strong and it was easy for all reviewers to understand it throughly. I believe researchers who study data efficient RL will benefit a lot from this paper and will build on top of it.

There were some minor concerns about novelty, perhaps due to the simplicity of the approach but I think its simplicity is a strength. I would not have expected transformers to work as well as they do in such a setting on the 100k Atari problem. So it is definitely a surprising and interesting result.

**Note From Pc:**

if the above contains the word "oral" or "spotlight" please see: "oral" presentation means -> notable-top-5% and "spotlight" means -> notable-top-25%. As stated in our emails, we are disassociating presentation type from AC recommendations